# Identification of barriers, facilitators and system-based implementation strategies to increase teleophthalmology use for diabetic eye screening in a rural US primary care clinic: a qualitative study

Yao Liu,[1,2] Nicholas J Zupan,[1] Rebecca Swearingen,[1,2] Nora Jacobson,[3] Julia N Carlson,[1] Jane E Mahoney,[4] Ronald Klein,[1] Timothy D Bjelland,[5] Maureen A Smith[2,6]

For numbered affiliations see end of article.

**Correspondence to**
Dr Yao Liu; liu463@wisc.edu, yaoliumd@gmail.com

## ABSTRACT

**Objective** Teleophthalmology for diabetic eye screening is an evidence-based intervention substantially underused in US multipayer primary care clinics, even when equipment and trained personnel are readily available. We sought to identify patient and primary care provider (PCP) barriers, facilitators, as well as strategies to increase teleophthalmology use.

**Design** We conducted standardised open-ended, individual interviews and analysed the transcripts using both inductive and directed content analysis to identify barriers and facilitators to teleophthalmology use. The Chronic Care Model was used as a framework for the development of the interview guide and for categorising implementation strategies to increase teleophthalmology use.

**Setting** A rural, US multipayer primary care clinic with an established teleophthalmology programme for diabetic eye screening.

**Participants** We conducted interviews with 29 participants (20 patients with diabetes and 9 PCPs).

**Results** Major patient barriers to teleophthalmology use included being unfamiliar with teleophthalmology, misconceptions about diabetic eye screening and logistical challenges. Major patient facilitators included a recommendation from the patient's PCP and factors related to convenience. Major PCP barriers to referring patients for teleophthalmology included difficulty identifying when patients are due for diabetic eye screening and being unfamiliar with teleophthalmology. Major PCP facilitators included the ease of the referral process and the communication of screening results. Based on our results, we developed a model that maps where these key patient and PCP barriers occur in the teleophthalmology referral process. Patients and PCPs also identified implementation strategies to directly address barriers and facilitators to teleophthalmology use.

**Conclusions** Patients and PCPs have limited familiarity with teleophthalmology for diabetic eye screening. PCPs were expected to initiate teleophthalmology referrals, but reported significant difficulty identifying when patients are due for diabetic eye screening. System-based implementation strategies primarily targeting PCP barriers in conjunction with improved patient and provider education may increase teleophthalmology use in rural, US multipayer primary care clinics.

## Strengths and limitations of this study

► We used qualitative methods to capture the real-world perspectives of patients and providers regarding barriers and facilitators to teleophthalmology use in a rural US multipayer primary care clinic with an active teleophthalmology programme.

► We identified and categorised implementation strategies directly suggested by patients and providers using the Chronic Care Model.

► All patients were Caucasian and native English speakers.

► Most patients in this study self-reported high levels of general health literacy (85%), which was greater than that reported by rural adults from a similar population (70.9%).

► We did not systematically assess patient knowledge of diabetic eye screening.

## INTRODUCTION

There are an estimated 4.2 million Americans with diabetic retinopathy, which is the most common cause of blindness in working-age US adults.[1 2] The risk of severe vision loss decreases by 90% with early diagnosis and treatment, but fewer than half of the 29.1 million Americans with diabetes receive yearly recommended diabetic retinopathy screening.[3–5] Teleophthalmology is an evidence-based intervention proven to substantially improve diabetic eye screening rates and reduce blindness from diabetes.[6] A retinal camera is used to image patients' eyes in a convenient location, such as a

primary care clinic (where more than 90% of patients with diabetes obtain their care).[7] These images are then electronically transmitted to and evaluated by specialists at a distant site, typically within a time frame of the same day to 1 week. Patients needing additional eye care are then identified for expedited treatment. The prevalence of diabetes and the demand for eye screening is projected to double by 2050 without a concurrent increase in the supply of eye care providers.[8] Thus, there is an urgent need to expand teleophthalmology use to improve screening rates and respond to growing demand.

In England, the National Health Service achieves screening rates of over 80% using teleophthalmology and subsequently, diabetic retinopathy is no longer the leading cause of blindness in working-age English adults.[6 9] Presently, successful implementation of teleophthalmology in the USA is largely limited to single-payer or highly specialised health systems.[10 11] Teleophthalmology programmes in these settings have achieved sustained screening rates as high as 80% or more. Success in multipayer settings has been much more limited. A multipayer health system is one in which individuals (or their employers) pay for healthcare services through a variety of private or public health insurance sources, in contrast to a single-payer health system in which healthcare is paid for by a single payer (eg, government-financed healthcare supported by taxes).[12] A recent 5-year randomised controlled trial in a multipayer health system compared teleophthalmology to traditional screening methods (ie, in-person dilated eye examinations) and found initial improvement in screening rates with teleophthalmology, but screening rates declined within 18 months and did not exceed 55% even when teleophthalmology became available to both groups.[13] However, this study did not use a systematic implementation approach designed to sustain the integration of this technology into the primary care workflow.

Teleophthalmology is particularly well suited to rural populations, which have less access and greater travel distances to obtain eye care.[14–18] Rural communities are largely served by multipayer health systems, which are less likely to encourage preventive services because of poor reimbursement for such services due to insurers' financial incentives to focus on providing healthcare in the short term.[12] Unfortunately, studies show that simply providing access to teleophthalmology in multipayer health systems is insufficient to either achieve or sustain initial improvements in screening rates.[10 12 13 19 20] Patient and provider-related barriers to teleophthalmology have been postulated, but are poorly understood. These barriers may be magnified in rural populations because they are less insured, poorer, older, less likely to receive guideline-concordant care, and experience more chronic diseases than those in urban areas.[16–18 21]

We hypothesised that while teleophthalmology addresses some logistical barriers to diabetic eye screening, it may not address other important patient and provider barriers in multipayer health systems. Qualitative research methods are used in health services research to explore complex phenomena needing further explanatory analysis, such as real-world patient and provider barriers to teleophthalmology use, through a rich description of key perspectives.[22] We conducted individual interviews to understand what prevents or motivates patients and primary care providers (PCPs) to use teleophthalmology, as well as strategies to increase teleophthalmology use, in a rural multipayer health system with an active teleophthalmology programme.

## RESEARCH DESIGN AND METHODS
### Research setting
We conducted standardised open-ended, individual interviews with patients with diabetes and PCPs at Mile Bluff Medical Center. Mile Bluff is a rural, multipayer health system in Mauston, Wisconsin, USA. A teleophthalmology programme was established in 2015 (1 year prior to the start of our study) in partnership with the University of Wisconsin (UW)-Madison. This programme allows PCPs to refer patients for teleophthalmology with walk-in scheduling. Referrals are completed by PCPs in the Mile Bluff electronic health record (EHR). Retinal images are electronically transmitted to and evaluated by university eye specialists. Imaging reports are then sent back to the PCP and patient within 1 week, which is consistent with the usual time frame for receiving results of other clinical studies (eg, laboratory tests and X-rays) provided by this rural health system and was considered acceptable to all patients and PCPs in our study. Patients are referred to local eye doctors for further care if found to have visually significant eye disease.

This teleophthalmology programme was established prior to our study based on the 2011 American Telemedicine Association Telehealth Practice Recommendations for Diabetic Retinopathy.[23] The Topcon NW400 camera (Topcon Medical Systems, Oakland, New Jersey, USA) was used to obtain a single 45° image of the disc and macula in each eye, along with an anterior segment photograph. If a fundus image was considered to be poor quality by the imager, then the camera's 'small pupil' mode was used to capture additional images. If images remained poor, then pharmacological dilation using 0.5% tropicamide was performed with the patient's consent. The percentage of patients undergoing pharmacological pupil dilation was 2.2% and the frequency of ungradable cases among all patients was 2.6%. Two years after the establishment of the teleophthalmology service, a quality improvement programme outside the scope of this study was created to develop an implementation programme to increase teleophthalmology utilisation.

### Interviews
We developed our interview guides using the Chronic Care Model, a framework for improving chronic disease management and guideline-concordant diabetes care.[24 25] Following a literature search on barriers and facilitators to teleophthalmology for diabetic eye

screening, the patient interview guide (online supplementary appendix S1) was tested and further refined with input from the UW Community Advisors on Research Design and Strategies (CARDS), which is a group of lay community members trained to review patient research materials. The PCP interview guide (online supplementary appendix S2) was tested and further refined with input from PCPs in the UW Primary Care Academics to Transform Healthcare. Members of the UW Institute for Clinical and Translational Research-Community Academic Partnership (UW ICTR-CAP) Qualitative Research Group also reviewed and provided feedback on both interview guides.

Standardised open-ended, individual interviews combined with flexible probes were conducted between July 2016 and April 2017 (1–2 years after the teleophthalmology programme was established) to understand patient and PCP perspectives on teleophthalmology and diabetic eye screening. This approach to interviewing ensured that each interview covered all topics included in the guide consistently, but also allowed the interviewer latitude to explore specific participant responses.[26] All interviews were conducted one on one by a female research specialist (RS) with training in qualitative research and certification as a nursing assistant. The interviewer did not have a relationship with the participants prior to the study and informed participants that she did not have specialised medical training in eye care. Patient interviews were conducted in person (30–45 min) at a local library. PCP interviews were conducted over the phone (15–30 min) to accommodate their busy clinic schedules. Patients received US$30 and PCPs received US$50 compensation for their time. Interviews were audiotaped and transcribed verbatim with all personal identifiers redacted. The interviewer also took field notes during and after the interview.

### Study sample
The sample of 20 adult patients with diabetes (with and without experience with teleophthalmology) and 9 PCPs was drawn from patients and providers at Mile Bluff Medical Center. Adult patients (18 years or older) with a diagnosis of diabetes were recruited who either: (1) had teleophthalmology imaging within the preceding 2 months or (2) expressed interest in participating in a research study when previously contacted in a quality improvement telephone survey on diabetic eye screening. Fifty patients were invited to participate by a mailed letter and a follow-up phone call. All patients had a PCP from Mile Bluff Family or Internal Medicine. PCPs were recruited during a provider staff meeting with purposeful recruitment of providers having a range of training backgrounds reflective of their representation at Mile Bluff. No participants dropped out of the study. Sample sizes for both patients and PCPs were sufficient to reach informational redundancy in which no new information was obtained from additional interviews.[27]

### Data analysis
We began analysing the interview transcripts using inductive analysis. Members of the research team (RS-research specialist and certified nursing assistant, NJZ-research specialist and YL-clinical ophthalmologist and principal investigator) conducted independent open coding of the first five transcripts. Research team members then met with NJ, qualitative methodologist, to review these codes and agree on an initial coding framework. A second coding cycle was then performed by one research team member (NJZ) to fit codes into an evolving collection of higher order categories. Consistency was ensured by the principal investigator (YL) who dual coded every fifth transcript. Throughout the analysis process, codes were iteratively reviewed by the entire research team, which met regularly to discuss and refine the first and second-order analytical categories pertinent to understanding facilitators and barriers to teleophthalmology use.

Research team discussions centred around the organisation of themes into separate categories (eg, lack of time) or grouping them within larger categories (eg, logistical challenges), as well as how themes applied to patients, PCPs or to both groups. The team also used these meetings to explore other, less expected, themes emerging from the data. Unexpected findings included the influence of broader socioecological factors on rural residents' adherence with diabetic eye screening, which were beyond the scope of this study.[28] These included limited access to healthcare (eg, long travel distances), anxiety stemming from family members' experiences with diabetes complications and the daily burden of diabetes management. Additionally, implementation strategies to increase teleophthalmology use were deductively coded and categorised using the Chronic Care Model as a framework for directed content analysis. Members of the UW ICTR-CAP Qualitative Research Group also reviewed parts of the interview data and coding methods. We used NVivo software, V.11.4.1 (QSR International, Melbourne, Australia) for data management.

Our data analysis was validated using member checking with a subset of interview participants (n=9 patients, n=6 PCPs) at two separate patient and provider stakeholder group meetings organised as part of a quality improvement initiative to increase diabetic eye screening at Mile Bluff that began 2 years after the establishment of the teleophthalmology programme to increase its utilisation.[29] Participating patients and providers judged our interpretation of the interview data to be accurate and complete. Furthermore, the most important or 'top' barriers and facilitators were identified through the nominal group technique, in which each participant sequentially shares one additional idea with the group (until no new ideas are generated) and then all participants anonymously cast their written votes.[30] Our report of this study followed the Consolidated Criteria for Reporting Qualitative Research.[31]

## Patient involvement

Patients from the UW CARDS were involved in the design of the study by providing feedback on the development of the patient interview guide. In addition, a patient stakeholder group (Mile Bluff Diabetes Patient Advisory Council) was established among participants in this study to provide member checking of our results and advise on dissemination of the results.

## RESULTS

### Patient and provider characteristics

All patients were Caucasian adults diagnosed with type 2 diabetes (table 1). Fifty per cent of patients had experience with teleophthalmology and most (85%) self-reported high health literacy in response to the single-item literacy screener.[32] Only 1 patient among the 20 patient participants underwent pharmacologic pupil dilation during a teleophthalmology visit. There were no significant differences in the responses of this patient compared with those of the other 19 patients who did not undergo pupil dilation. PCPs were predominantly male (77.8%) and had a variety of training backgrounds (table 1). Most had been in practice for over 10 years (77.8%) and had referred patients for teleophthalmology (87.5%).

### Patient barriers and facilitators

Major patient barriers to teleophthalmology use included being unfamiliar with teleophthalmology, misconceptions about diabetic eye screening and logistical challenges (table 2).

Major patient facilitators included a recommendation from the patient's PCP and factors related to convenience. Most patients were unfamiliar with teleophthalmology, which prevented them from taking the initiative to seek a teleophthalmology referral from their PCP for diabetic eye screening. In addition, several patients demonstrated a limited understanding of diabetic eye disease and the importance of screening.

> I don't see an advantage to getting [my eyes] checked every year, unless you are having issues… (Patient 5)

Many patients expressed the belief that having 'good' vision and the absence of visual symptoms indicated that they do not have diabetic eye disease. The advantage of identifying and treating the disease at earlier, asymptomatic stages was not well understood. Patients were often unaware that ongoing screening was needed because the risk of retinopathy increases over time. Some patients believed that having one diabetic eye screening that was negative, even if it was several years ago, was sufficient and that additional screening was unnecessary.

Interestingly, neither a detailed understanding of diabetic eye disease nor the purpose of screening was necessary for patient adherence if the patient believed that yearly diabetic eye screening was important. This belief was often attributed to a strong recommendation

| Table 1 | Patient and primary care provider demographics |
| --- | --- |
| **Participant characteristics** | **Median or percentage** |
| Patients (n=20) | |
| Age | 67 years (range: 46–86 years) |
| Male | 55 |
| Ethnicity (self-reported) | |
| Caucasian, non-Hispanic | 100 |
| Diagnosis of type 2 diabetes | 100 |
| Had teleophthalmology screening | 50 |
| Duration of diabetes | |
| <5 years | 40 |
| 5–19 years | 30 |
| 20+ years | 30 |
| Highest level of education | |
| College graduate | 10 |
| Some college/tech school | 30 |
| High school graduate or General Education Diploma (GED) | 35 |
| Some high school | 15 |
| Grade 8 or less | 10 |
| Health literacy (single-item literacy screener) | |
| High | 85 |
| Moderate | 10 |
| Low | 5 |
| Primary care providers (n=9) | |
| Male | 77.8 |
| Training background | |
| Physician | 44.4 |
| Physician Assistant-Certified (PA-C) | 33.3 |
| Doctor of Nursing Practice (DNP) | 11.1 |
| Registered Nurse (RN) | 11.1 |
| Years in practice | |
| >10 years | 77.8 |
| 5–10 years | 0 |
| 0–5 years | 22.2 |
| Have referred patients for teleophthalmology | 87.5 |

from their PCP, which was the most common reason patients reported for using teleophthalmology.

> My doctor thinks [teleophthalmology], you know, is a good test. And for me that's pretty much all I need. (Patient 16)

A PCP's recommendation was a major patient motivator because of the high level of trust patients placed in

**Table 2**  Patient and primary care provider barriers and facilitators

| Patients | Barriers | ▶ Unfamiliar with teleophthalmology.* |
|---|---|---|
| | | ▶ Misconceptions about diabetic eye screening.* |
| | | ▶ Logistical challenges* (eg, time, transportation, out-of-pocket cost). |
| | | ▶ Eye problems requiring in-person examination (eg, glasses or glaucoma). |
| | | ▶ Anxiety about receiving bad news regarding their eyes. |
| | Facilitators | ▶ Recommendation from primary care provider.* |
| | | ▶ Convenience of teleophthalmology* (eg, same-day scheduling, location, quick). |
| | | ▶ Belief that diabetic eye screening is important for preventing vision loss. |
| | | ▶ Knowing that pharmacologic pupil dilation is usually not necessary. |
| | | ▶ Teleophthalmology is considered a high-quality service due to University affiliation. |
| Primary care providers | Barriers | ▶ Difficulty identifying when patients are due for diabetic eye screening.* |
| | | ▶ Unfamiliar with teleophthalmology.* |
| | | ▶ Time constraints (eg, many competing tasks during clinic visit). |
| | | ▶ Concerns about conflicts with local eye doctors. |
| | | ▶ Concerns about patients' barriers (eg, out-of-pocket cost). |
| | Facilitators | ▶ Ease of referral process and results communication.* |
| | | ▶ Perceived benefits to patients (eg, convenience, cost). |
| | | ▶ Improved patient adherence with diabetic eye screening. |
| | | ▶ Benefits to the healthcare organisation (eg, increased reimbursement for improved quality metrics). |

*Top barrier or facilitator identified at patient or primary care provider stakeholder meeting.

their PCP. Patients reported that they obtained most of their health information directly from their PCP as well as through the clinic's diabetes education programmes and informational handouts. Some patients also described maintaining independence, continuing to enjoy hobbies and caring for other family members as important reasons for protecting their vision through screening. Their PCP's recommendation to pursue teleophthalmology screening reinforced these personal values.

Teleophthalmology was frequently endorsed as being 'quick, easy and painless' (patient 5). When comparing teleophthalmology to traditional, in-person eye examinations, many patients appreciated that teleophthalmology was conveniently located in the same building as their PCP and accommodated walk-in scheduling with short wait times. In addition, they described the imaging as highly efficient, often taking 'less than 5 min' (patient 19). Teleophthalmology was often preferred by patients because it was more comfortable than traditional eye examinations since pharmacological pupil dilation was usually not needed. Pharmacological pupil dilation was reported as a significant barrier to obtaining traditional eye examinations.

Look at it from my viewpoint. I'm upstairs at the doctor's office and he says, 'Maybe you should go get [an eye photo] done…' You walk downstairs… sit down for 10 min… you get it done and you go home… I mean, it's so simple. (Patient 17)

The last thing I want to do is lose my eyesight [from diabetes] and before [teleophthalmology], I wasn't big on going to the eye doctor… (Patient 16)

I have a terrible time when they dilate my eyes. The light, it hurts so bad. (Patient 5)

Although teleophthalmology addresses many barriers to diabetic eye screening, several important challenges remain for patients in this older, rural population with diabetes, many of whom live on a limited income. Teleophthalmology is only available in one primary care clinic so patients at other clinic locations were required to travel from their regular clinic to use it (median travel distance: 13.8 miles, range: 9.2–23.1 miles). Most patients also reported the need to pay out of pocket as a barrier due to limited insurance coverage for teleophthalmology.

### PCP barriers and facilitators
Major PCP barriers to teleophthalmology use included difficulties identifying when patients are due for diabetic eye screening and being unfamiliar with teleophthalmology, while major facilitators were the ease of the referral process and results communication (table 2). PCPs reported that they often did not have access to patients' eye records since all eye care providers in this community practice outside their health system and use different EHRs. Thus, PCPs depend on eye care providers to send diabetic eye screening documentation, which is not consistently performed. PCPs reported insufficient reminders, time and resources to 'track down' eye records as well as to discuss and refer patients for teleophthalmology as significant barriers. For example, enlisting the help of a medical assistant to request records from the patient's eye doctor was difficult due to time constraints during a typical PCP appointment. When a PCP cannot easily find these records, they would usually rely on the patient's self-reported date of their last diabetic eye screening, which has limited accuracy.

I don't have an easy way in my electronic records to see if [patients] had [diabetic eye screening] done and the patients never remember exactly when [it was done]… (PCP 7)

Faced with so many competing demands, PCPs often prioritised more urgent medical issues over teleophthalmology.

[As a provider] you are covering so many things… the foot exam, checking their cholesterol… [patients say] they saw the eye doctor and then you just go on to something else… (PCP 4)

PCPs also reported being unfamiliar with teleophthalmology, which made some hesitant to refer patients. Some PCPs also worried about potential conflicts with local eye doctors and preferred to refer patients to eye doctors whom PCPs felt would be better suited to judge whether teleophthalmology would be appropriate.

I guess I just don't want to refer [a patient] in error if it's not really what the service is meant for.' (PCP 3)

[I]… tend to refer patients to the ophthalmologists and they would probably be the ones to decide whether they were… teleophthalmology candidates (PCP 5).

The most important facilitators for PCPs were the ease of referral process and results communication.

[Teleophthalmology] was probably one of the easiest referrals I've done, and the turnaround [time for receiving results] was by far the best… I've had (PCP 3).

PCPs described teleophthalmology as being convenient for their patients and endorsed it for making it easier for PCPs to document referrals and reliably receive reports in their EHR among patients who had received diabetic eye screening. PCPs were also motivated to use teleophthalmology because they believed that this technology made it easier and more likely for patients to obtain diabetic eye screening. If patients did not have access to teleophthalmology, patients would otherwise need to make their own appointments with an eye care provider for a dilated eye examination to obtain diabetic eye screening, which

was more difficult due to the multiple patient barriers we described.

### A model to understand patient and PCP barriers in the teleophthalmology referral process

Based on our results, we developed a model for understanding where key patient and PCP barriers occur during the primary care teleophthalmology referral process (figure 1). This model illustrates the temporal relationships between barriers in the teleophthalmology referral process, which are important for understanding the relative impact and mapping of possible strategies to overcome barriers in this process. PCPs are expected to initiate the referral process, but have difficulty remembering to ask patients about diabetic eye screening and identifying when patients are due. Next, they may have limited time to generate the referral and explain its purpose to the patient, especially when both the patient and the PCP are less familiar with teleophthalmology. Finally, referred patients may choose not to participate in teleophthalmology due to being unfamiliar with the technology, misconceptions or a lack of understanding about the importance of diabetic eye screening and/or logistical difficulties related to time, travel or cost. This figure helps to demonstrate why strategies aimed at the provider and health system may be more effective in increasing teleophthalmology use because of the earlier role of the PCP and the later role of the patient in completing the referral process.

### Possible implementation strategies

Patients and PCPs described possible implementation strategies to increase teleophthalmology use, which were categorised using the Chronic Care Model (table 3). A variety of system-based implementation strategies were identified, primarily targeting the health system by streamlining PCP and clinic staff workflow processes (table 3).

Strategies that addressed top barriers or facilitators included best practice alerts in the EHR to remind PCPs when patients are due for diabetic eye screening, improving PCP access to diabetic eye screening records and clinic workflow changes such as the delegation of teleophthalmology referrals to clinic staff. In contrast, fewer patient-directed strategies were identified. Arranging the scheduling and location for teleophthalmology

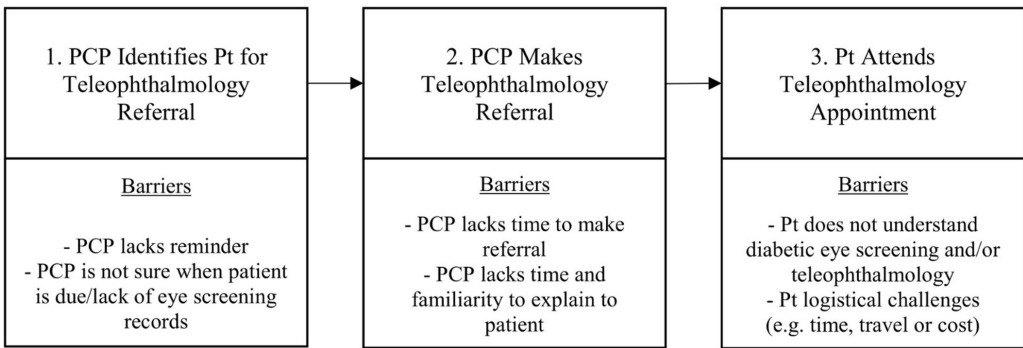

**Figure 1** Barriers in the teleophthalmology referral process. PCP, primary care provider; Pt, patient.

**Table 3** Strategies to increase teleophthalmology use mapped to Chronic Care Model (CCM)

| CCM component | Target | Examples of strategies |
|---|---|---|
| Health system organisation and delivery system design | Health system, PCPs and clinic staff | ► Workflow changes including clinic staff checklists and delegation of referrals.* <br> ► Provider/staff training to increase familiarity with teleophthalmology.* <br> ► Convenient scheduling and location.* <br> ► Provide financial incentives to individual PCPs for diabetic eye screening performance. |
| Decision support and clinical information systems | Health system, PCPs and clinic staff | ► Best practice alert in EHR when the patient is due for diabetic eye screening.* <br> ► Streamline processes for getting diabetic eye screening documentation into EHR.* <br> ► Provide PCPs with feedback/data on diabetic eye screening performance (eg, quarterly). <br> ► Generate lists of patients due for diabetic eye screening for clinic staff to contact. |
| Self-management support and community resources | Patients, families, community members and clinic staff | ► Patient education materials provided at primary care clinic visits.* <br> ► Increase education about diabetic eye screening in diabetes self-management classes.* <br> ► Publicise teleophthalmology services* (eg, local media and community health fairs). <br> ► PCP clinic staff facilitate diabetic eye screening by calling patients and sending letters when due. |

*Strategies that can address top barriers or facilitators from table 2.
EHR, electronic health record; PCP, primary care provider.

imaging to maximise patient convenience was emphasised. Suggested solutions included having teleophthalmology available during weekend or evening hours, as well as obtaining a camera for each primary care clinic or rotating the current camera between primary care clinics (eg, a 'mobile' camera approach). Many PCPs recommended providing further training to both providers and clinic staff on the use and purpose of teleophthalmology. Patients reported that greater education about the importance of diabetic eye screening, particularly to address common misconceptions, and increasing community awareness of teleophthalmology services would increase their likelihood of using teleophthalmology. Many patients preferred receiving written patient education materials from their PCP clinic or through presentations in diabetes education classes. They also recommended publicising teleophthalmology through the local newspaper or television advertisements.

## DISCUSSION

Teleophthalmology is effective for increasing diabetic eye screening rates in rural populations.[14–18] However, there may be poor early adoption of this technology or a lack of sustained use by patients and PCPs despite having both equipment and trained personnel readily available—and even when the service is provided at no cost to the patient.[13] In this qualitative study, we identified patient and provider barriers, facilitators and implementation strategies to improve and sustain teleophthalmology use over time for diabetic eye screening in a rural, multipayer primary care setting.

Top patient barriers include being unfamiliar with teleophthalmology, misconceptions about diabetic eye screening and logistical challenges, which were consistent with prior studies examining patient barriers to teleophthalmology and traditional methods for diabetic eye screening.[33–41] Patients reported that the most common reason they obtained screening was a strong recommendation from their PCP. Many prior studies that did not ask about provider recommendation have found that patients were primarily motivated to seek diabetic eye screening to prevent vision loss,[33 34 36] but this was less frequently reported by patients in our study. We found that while many patients have limited knowledge and understanding of the purpose of diabetic eye screening, they still obtained teleophthalmology screening when recommended by their PCP. Our data agree with literature showing that patients report provider recommendation as the strongest motivator of obtaining preventative screenings.[40 42] A survey of nearly 2000 patients with diabetes found provider recommendation to be the most strongly associated with patient adherence with diabetic eye screening (OR 341, 95% CI 164 to 715), and was much more strongly associated with adherence than patient knowledge about the effects of diabetic retinopathy on vision (OR 3.3, 95% CI 2.0 to 5.5).[40] We believe this is due

to patients' high level of trust in their providers, which has been linked to increased adherence with diabetic eye screening among rural older adults.[43]

In addition, our model demonstrates that the PCP initiates the teleophthalmology referral and that the patient's role in completing diabetic eye screening is often further downstream. The organisation of the clinic requires that the PCPs provide a referral for a patient to obtain teleophthalmology. As a result of this temporal relationship, if the PCP is not reminded to consider making the referral, the patient may not obtain teleophthalmology. Some strategies identified to address this barrier are the delegation of the teleophthalmology referrals to clinic staff such as medical assistants or allowing patient self-referral. Some PCPs reported concerns about the potential conflict between providing teleophthalmology in primary care and reduced referrals to local eye doctors, but studies have shown that teleophthalmology actually increases patient utilisation of local eye care services by bringing into care many patients who would not otherwise obtain yearly eye examinations.[44 45] While patient education is important, our model suggests that implementation strategies to address PCP barriers may be more effective for increasing teleophthalmology use because of the PCP's primary role in the referral process. This is supported by studies that showed no significant differences in demographics or beliefs among patients who are or are not adherent with diabetic eye screening.[36] Instead, our data suggest that PCPs and clinic staff characteristics may be more influential in determining whether patients adhere with screening.

System-level implementation strategies are needed to provide PCPs with the information needed to make appropriate referrals, including better communication with eye care providers (eg, up-to-date documentation of diabetic eye examinations) and other EHR enhancements (eg, systematic reminders when patients are due for screening and streamlining the referral process).[33 35] We recognise that the cost and availability of certain EHR functionalities are a potential limitation for some health systems and that lower cost alternatives or other implementation strategies may be more important in settings with limited resources. For example, one of the medical assistants in this primary care clinic devised a low-cost system of flagging paper charts with a colour-coded diabetes checklist to make it easier for the PCP identify patients due for diabetic eye screening. Addressing logistical barriers for patients by providing same-day, walk-in scheduling at a convenient location and minimising out-of-pocket costs are also important. In addition, PCP, clinic staff and patient education as well as publicising teleophthalmology throughout the community were also recommended by interview participants. Of note, we did not ask study participants to differentiate between strategies they believed would increase initial adoption or sustain the use of teleophthalmology over time. Determining which of the strategies they identified would be useful for one or both purposes should be evaluated in future studies.

There are several limitations of our study. All patients reported having experience with diabetic eye screening and may have been more willing to participate than patients who do not follow diabetes eye screening guidelines. However, studies show that similar barriers exist for patients who do and those who do not adhere with screening guidelines.[36] Our teleophthalmology programme requires PCP referral and different barriers may occur in teleophthalmology programmes allowing patient self-referral. The majority of patients in our study reported high levels of general health literacy (85%), which was greater than that reported by a similar population of rural adults (70.9%).[46] We did not systematically assess knowledge of diabetic eye screening from patients in our study because prior literature has demonstrated that more than half of patients with diabetes are aware of screening guidelines and patient knowledge of guidelines is insufficient to ensure adherence with diabetic eye screening guidelines.[35 47]

In addition, all patients were Caucasian and native English speakers, which reflected the composition of this community. A study of predominantly Latino and African-American patients in an urban, safety-net clinic focused on gaps in patient knowledge of diabetic retinopathy and screening, which prevented them from using teleophthalmology.[41] Therefore, that study emphasised patient education and patient-provider communication rather than other system-level strategies that may address patients' utilisation of teleophthalmology. The rural population in our study faces limited access to care, which is shared by many non-Caucasian populations, including those in underserved urban areas and in low- to-medium-income countries. While we expect some of the implementation strategies we identified to translate to these populations, tailoring of strategies to the local community may be important to account for differences in cultural backgrounds and available healthcare resources. Further studies among patients from urban, other ethnic groups and non-native English-speaking backgrounds are needed to assess the generalisability of our findings.

Our study is one of the first to directly identify implementation strategies suggested by patients and PCPs that address their barriers and facilitators to using teleophthalmology for diabetic eye screening. In this rural, multipayer health system, PCPs initiate teleophthalmology referrals and patients reported that a strong recommendation from their PCP was the most important motivator for obtaining screening. Thus, system-level implementation strategies focusing on the PCP and clinic staff workflow processes appear to have the greatest potential to increase and maintain utilisation in this setting. Future studies testing these strategies in multiple rural and urban health systems can evaluate their relative impact and the generalisability of our findings on increasing teleophthalmology use and diabetic eye screening rates. Furthermore, interviews with health system administrators may be useful for identifying additional barriers to establishing and sustaining teleophthalmology programmes. Implications for other clinics include the importance of

strategies targeting health system workflow processes along with educating patients, providers and staff. Clinics may benefit from assessing their own unique patient and provider barriers and facilitators, as well as their available resources, in order to tailor the selection of effective strategies to increase teleophthalmology utilisation.

## CONCLUSIONS

Teleophthalmology can substantially increase diabetic eye screening rates and prevent blindness. While this technology addresses numerous logistical barriers to diabetic eye screening, we found several additional barriers to its use by patients and PCPs. System-based implementation strategies primarily targeting PCP barriers in conjunction with improved patient and provider education may increase teleophthalmology use in rural, multipayer primary care clinics.

**Author affiliations**
¹Department of Ophthalmology and Visual Sciences, University of Wisconsin-Madison School of Medicine and Public Health, Madison, Wisconsin, USA
²Health Innovation Program, University of Wisconsin-Madison School of Medicine and Public Health, Madison, Wisconsin, USA
³Institute for Clinical and Translational Research, University of Wisconsin School of Medicine and Public Health, Madison, Wisconsin, USA
⁴Department of Medicine, University of Wisconsin School of Medicine and Public Health, Madison, Wisconsin, USA
⁵Mile Bluff Medical Center, Mauston, Wisconsin, USA
⁶Departments of Population Health Sciences, Family Medicine and Community Health, University of Wisconsin School of Medicine and Public Health, Madison, Wisconsin, USA

**Acknowledgements** The authors acknowledge the UW ICTR-CAP Qualitative Research Group (QRG), Wisconsin Network for Research Support (WINRS), Community Advisors on Research Design and Strategies (CARDS) and the Primary Care Academics Transforming Healthcare (PATH) Writing Collaborative for their feedback on interview guides and recruitment methodology. We thank the Mile Bluff Medical Center Diabetes Patient Advisory Council and Diabetes Quality Care Team for providing feedback to refine our research results. Prior abstract publication: This study was presented at the 2017 Association for Research in Vision and Ophthalmology (ARVO) Annual Meeting in Baltimore, Maryland, USA.

**Contributors** YL, RS and NJ developed the interview guides. RS conducted the interviews. YL, NJZ, JNC and NJ analysed the data. JEM, RK, TDB and MAS contributed to the discussion and reviewed/edited the manuscript. YL and NJZ wrote the manuscript.

**Funding** This work was supported by NIH/NEI K23 EY026518-02 and a Wisconsin Partnership Program New Investigator Award. It was also supported, in part, by an institutional grant from Research to Prevent Blindness, New York City, New York, USA to the University of Wisconsin School of Medicine and Public Health, Department of Ophthalmology and Visual Sciences. Additional support came from the Clinical and Translational Science Award (CTSA) program, through the NIH National Center for Advancing Translational Sciences (NCATS), grant UL1TR000427. Further support was received from the University of Wisconsin (UW) School of Medicine and Public Health Innovation Program, the Wisconsin Partnership Program and the UW Institute for Clinical and Translational Research-Community Academic Partnerships (ICTR-CAP) core.

**Disclaimer** The study sponsors had no role in the design and conduct of the study; collection, management, analysis and interpretation of the data; preparation, review or approval of the manuscript; and the decision to submit the manuscript for publication. The content is solely the responsibility of the authors and does not necessarily represent the official views of the NIH.

**Competing interests** None declared.

**Patient consent for publication** Not required.

**Ethics approval** The University of Wisconsin School of Medicine and Public Health Human Subjects Institutional Review Board (IRB) staff reviewed all study activities in detail and determined that this research met criteria for exemption from full IRB review based on US federal Common Rule (45 CFR 46.101(b)), which provides exemptions for interview research protocols with minimal risk to participants.

**Provenance and peer review** Not commissioned; externally peer reviewed.

**Data sharing statement** Data from this study are available on request from the corresponding author to those who fulfill requirements set by the University of Wisconsin Institutional Review Board.

**Author note** YL accepts full responsibility for this work as a whole, including the study design, access to the data and the decision to submit and publish the manuscript.

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
