## [Reviewer comments · BMJ Open]

ARTICLE DETAILS

TITLE (PROVISIONAL)	Identification of Barriers, Facilitators, and System-based Implementation Strategies to Increase Teleophthalmology Use for Diabetic Eye Screening in a Rural U.S. Primary Care Clinic: A Qualitative Study
AUTHORS	Liu, Yao; Zupan, Nicholas; Swearingen, Rebecca; Jacobson, Nora; Carlson, Julia; Mahoney, Jane E.; Klein, Ronald; Bjelland, Timothy; Smith, Maureen

VERSION 1 – REVIEW

REVIEWER	Lil Deverell Swinburne University of Technology, Australia
REVIEW RETURNED	27-Jun-2018

GENERAL COMMENTS	Congratulations to the authors on conducting a study that helps to close the gap in translational research. There seems little value in having fabulous technology to avoid preventable blindness if we don't understand the factors affecting rejection or uptake of this technology. This article makes a useful contribution to this body of knowledge. I recommend that the article be accepted for publication after response to the following considerations. Page 3 line 24: I suggest you replace "performed" with "conducted" in relation to interviews throughout the manuscript. Interviews involve a two-way conversation, whereas performance is one-way and doesn't require a response. This semantic distinction points to the shift from researcher power and the assumption of objectivity in quantitative or positivist research to shared power and a constructivist approach to knowledge (exploring multiple meanings and interpretations to build knowledge together) that is characteristic of good qualitative research. Page 5, line 22: the last point in the article summary seems to represent unexamined results that need further analysis and discussion to be meaningful in this diabetic retinopathy (DR) context. Thus, the article summary might include a statement like: "Despite 85% of patients reporting high level health literacy, their behaviour indicated dependence on their PCP to initiate regular screening for diabetic retinopathy." This statistic (85%) comes from a single non-specific question in the interview schedule, and could be examined in relation to the interview question on S1, page 1, line 47ff – "Some people get yearly eye checks..." as well as S1, page 6 question 6. If there are no adequate data from these questions to explore patients' knowledge about diabetic retinopathy (DR), then consider it a limitation of the study. Consider: Were patients aware of the DR guidelines for annual screening? If so, where did they learn about these guidelines? Has awareness of the DR guidelines changed their sense of responsibility around managing their diabetes and the
---

	risk of DR? If they haven't heard of the guidelines, what do they think would be the best places to learn about them? What would help patients to share responsibility for their DR screening, rather than leaving it up to the PCP? Page 7, line 24: I wondered what you meant by "rapidly"? Later in the article it becomes apparent that it takes a week for this particular clinic to get results, but it would be helpful at this point to give a typical time range for results from retinal photos to come back. Page 7, line 47: What does "multi-payer setting" mean? Can you add a sentence explaining this for readers outside the US please? Who are the payers and what proportion do they each contribute? This phrase was important enough to include in the article title and just needs some cultural definition. Page 8, line 15: Why are multi-payer health systems less likely to encourage preventive services? This issue of professional responsibility warrants further discussion, consideration of possible solutions and investigation in future studies. Page 8, line 26: Is Reference 22 the best example of guidelines for DR management? If so, it is worth referencing them here. Page 8, line 31: I would make hypothesize past tense, ie, hypothesized, but then I question the certainty of a hypothesis as an introduction to a qualitative study. It can be less prescriptive and more open to participants' lived experience to simply state a research question. Either way, you have done a good job of making your purpose clear in this study. Page 9, line 8: I wondered here when the study occurred, and suggest bringing forward the time-frame from page 10, line 22 to read "...at Mile Bluff Medical Centre, 1-2 years after the teleophthalmology program was established." Page 9, lines 29-36: the comments around deliberately NOT tailoring an implementation program for the teleophthalmology service raised ethical questions for me about withholding best services from patients, and wise stewardship of resources – investing in expensive medical technology then not using it to best advantage. From an objective research perspective, I can understand your reasoning in not promoting the program, so that a promoted program can be compared with an unpromoted program, but this was not the purpose of this study and neither were comparisons made in this study with another similar but promoted service. Page 9, line 42: The interview schedule seems to have been more structured than semi-structured. There was little evidence of the interviewer's willingness to depart from the questions to follow the participants' conversational cues about what was important to them in relation to DR. Similarly, there was little evidence in the results of the kind of surprises that tend to come from semi-structured interviews. While a structured interview schedule makes it easier to manage and analyse data in an orderly fashion, it also precludes the opportunity to explore unexpected meanings and reach a greater depth of understanding of the case under investigation. A structured interview might seem more scientific and appealing to readers used to quantitative research, but In future qualitative studies I would encourage the interviewers/ research team to be a little less prescriptive and a little more open to exploring the participants' leads, cultural curiosities and context. These details can offer greater insight into participants' behaviours and attitudes in this context and subsequently resonate with other similar contexts. Page 11, lines 15-45: The research methods are stated clearly enough to be replicated, which is good. However, the research team seems to have rushed to data reduction without much deep analysis of meaning. The coding framework was determined after only 5
--	---

interviews, with an emphasis on showing accord between coders, rather than exploring the different interpretations that might be evident between coders from different professional backgrounds. Good qualitative research delays data reduction in order to explore meanings and interpretations as fully as possible before coming to conclusions (Richards, L. (2009). *Handling qualitative data: A practical guide* (2 ed.). London, England: Sage.) A few sentences in the Discussion section (beginning page 21) would be welcome, exploring some of these alternative perspectives available from the research team.

Page 11, line 52 – this quality improvement initiative is interesting, especially in light of the comment on page 9, line 33 about “natural history”. At what point did the initiative begin, and what shape did it take? It would be good to explain this more in the discussion section.

Page 12, line 5: the mention of Nominal Group Technique was a welcome surprise – good to see member checking included and multiple points of engagement with participants in the study.

Page 12, line 38: what was the study exempt from, and why? This is a little disquieting given my earlier ethical concerns.

Page 13, line 42-47: This table gives a good overview of your participants. It would be helpful to word the PCP roles in full, rather than using initials.

Page 14: Table 2 provides a useful indication of the steps in your thinking during analysis, making the process transparent. The paper would be even richer if you included some sample quotes in a fourth column, indicating the kinds of comments that fed into these categories.

Page 14, line 47: It is not quite clear to me how “improved patient adherence with diabetic eye screening” is a facilitator for PCPs. How was improved patient adherence manifest? Do you mean that the PCP and patient share the responsibility for making the screening happen and that PCPs welcome this sharing? Did some patients initiate their DR screening appointments? Please explain in the section on PCP barriers and facilitators on Page 17.

Pages 17-19: You have used a good selection of quotes to illustrate your observations.

Page 19, lines 19-43. This section seems redundant, and along with Figure 1 could be deleted from the article. I appreciate that the model might have helped you move your analysis forward, like notes scratched on the back of an envelope, but it doesn't add anything to the results, insights from, or rigour of your study. This model is surpassed by the more robust Table 3 which is clearly stated and shows succinctly how your results and processes fit with an existing credible model that informed your questions in the first place.

Page 21, line 21: I suggest you rename this section Discussion and invest in deeper discussion on the issues already mentioned. You seem to accept without question that the PCPs should be in charge of prompting DR screening because this is the current assumption. Is there not a shift in US medical culture towards patient centred care and shared responsibility for health decisions, as there is in Australia, UK, etc? I would like to see some consideration of the local culture where the clinic is located. What makes people so dependent on their PCP, despite 85% being knowledgeable about health issues?

Page 22, Line 13: I suggest you reconsider this sentence, and perhaps combine it with the subsequent one, e.g. “In this community it seems that the PCP initiates the teleophthalmology referral, so if the PCP is not reminded to consider making the referral...” is this about organisational processes within the clinic, or is it about professional responsibility and power? Is it worth considering task

	shifting vs task sharing, relating to professional power/responsibility/status (see Shah et al 2018, "Task sharing in the eye care workforce: Screening, detection, and management of diabetic retinopathy in Pakistan. A case study"). This issue may or may not be relevant to your context, but it is worth discussing as a team. Also, are there promotional strategies that would encourage patients to remember their annual DR screening, such as linking it with their birthday, or a significant annual event in the community? Page 23, line 6: Your discussion of limitations of the study seems to begin here with "Of note..." but the issue you discuss from line 15 doesn't seem of great import, whereas a more detailed exploration of patient's health literacy has been omitted (either from lack of relevant interview data or lack of analysis?) Page 24, line 8: You refer generally to larger studies needed, but I would like some more specific suggestions about the most immediate research questions that have arisen from your study and how these questions might be investigated. Page 24, line 15: I suggest you reserve your Conclusion subheading for this paragraph, which neatly summarises your findings. Well done. Finally, a pleasing thing about this study is its very specific context – let it speak for itself. Then you might suggest some specific elements from this learning that will be transferrable to other contexts. What are the implications of your study, not just for this medical clinic but for others tasked with the responsibility for DR screening to prevent avoidable blindness? I enjoyed reading such a well-structured and well-phrased report.
--	--

REVIEWER	Sven-Erik Bursell University of Hawaii. USA
REVIEW RETURNED	28-Aug-2018

GENERAL COMMENTS	The authors present a useful study aimed at identifying patient and primary care provider (PCP) barriers and facilitators, as well as strategies to increase teleophthalmology use using a formalised approach. The manuscript is clearly written and the questionnaire data is well summarized so as to provide meaningful possible solutions to implementation and clinical operation. As a general comment the authors indicate that teleretinal imaging studies were transmitted to an external reading center with reports coming back to the primary care clinic within a week. This seems to be a long turn-around time and the patient may be lost to further interaction, especially in rural and underserved communities in low and middle income countries. The clinical workflow that has been most attractive is an initial retinal evaluation by the trained retinal imager at the point of encounter to determine whether or not that patient needs to be scheduled for specialty referral. Specific comments as follows:  1. It would be useful for the authors to provide more detail on the clinical and operational components of the teleophthalmology program, as these may impact on the responses provided by the patients. For example, what retinal imaging system was used, what imaging protocol was implemented and in general what percentage of all patients underwent pharmacological pupil dilation and what was the frequency of ungradable cases in this primary care population, how many patients a day underwent teleophthalmology screening (an indicator of imager proficiency). 2. As a corollary to the above it would be useful to understand how many of the 20 enrolled patients had to undergo pupil dilation and whether or not their response were different from those obtained
---

	from patients not having to undergo pupil dilation. 3. The authors indicate that one of the patient barriers was related to the availability of the service at their local primary care clinic. Did the authors consider a more mobile approach such as in the UK NHS program where cameras were moved from one clinic to another. A more mobile network would provide a useful solution here 4. In Table 3 the authors indicate that decision support and clinical information systems as a Chronic Care Model-based strategy. Unfortunately, despite the utility of such decision support and clinical information availability, there is a large level of inertia from Electronic Health Record vendors to provide this level of functionality. This is probably a result of EHR design philosophy, namely a primary focus on documenting elements to improve service reimbursement. So this strategy is fraught with barriers and may be too costly for a primary care user to afford. 5. The authors recognize that the patient population is Caucasian, most with high school or greater education, and with a high health literacy level. It would be interesting for the authors to provide some view of translation of these strategies into non-Caucasian populations or populations where resources were not as readily available.
--	---

VERSION 1 – AUTHOR RESPONSE

Reviewer #1

Response to General Comment #1: We appreciate Reviewer #1's positive remarks on the importance of our study in helping to close the gap in translational research.

2. Page 3 line 24: The word “performed” has been replaced with “conducted” everywhere that it appears in relation to the interviews.

3. Page 5 line 22: We have clarified in the Article Summary that health literacy refers to general health literacy and that we did not systematically assess patient knowledge of diabetic eye screening. We have now noted in the Discussion that we chose to focus our interview on patient beliefs and perceptions rather than knowledge assessment because prior literature has demonstrated that more than half of patients with diabetes are aware of screening guidelines and that patient knowledge of guidelines is insufficient to ensure adherence with diabetic eye screening guidelines. (Hartnett ME, Key IJ, Loyacano NM, Horswell RL, Desalvo KB. Perceived barriers to diabetic eye care: qualitative study of patients and physicians. *Arch Ophthalmol.* 2005;123(3):387–91. doi:10.1001/archophth.123.3.387.; Weiss DM et al. Effect of Behavioral Intervention on Dilated Fundus Examination Rates in Older African American Individuals With Diabetes Mellitus: A Randomized Clinical Trial. *JAMA Ophthalmol.* 2015 Sep;133(9):1005-12. doi: 10.1001/jamaophthalmol.2015.1760.)

4. Page 7 line 24: The typical time range for teleophthalmology image interpretation has been added in the Introduction.

5. Page 7 line 47: A multi-payer health system is one in which individuals (or their employers) pay for their health care services through a variety private or public health insurance sources, in contrast to a single-payer health system in which healthcare is paid for via a single payer (e.g. government-financed healthcare supported by taxes). This has been added to the Introduction.

6. Page 8 line 15: Multi-payer health systems in the U.S. are less likely to encourage preventative healthcare services because of poor reimbursement for such services due to insurers' financial

incentives to focus on providing healthcare in the short-term. This has been added to the Introduction.

7. Page 8 line 26: A reference to the American Diabetes Association guidelines for diabetic eye screening has been added. (10. Microvascular complications and foot care. *Diabetes Care*.2017;40(Suppl 1):S88–s98. doi:10.2337/dc17-S013.)

8. Page 8 line 31: “Hypothesize” has been changed to “hypothesized.”

9. Page 9 line 8: We clarified that the teleophthalmology program was established one year prior to our study.

10. Page 9 line 29-36: We have clarified in the manuscript that the teleophthalmology service was established prior to our study. Subsequently, a quality improvement program outside the scope of this study was created to develop an implementation program to increase teleophthalmology utilization.

11. Page 9 line 42: We have revised the Methods to clarify that we used a combination of standardized open-ended interviewing, a variant of semi-structured interviewing, (Patton, *Qualitative Research and Evaluation Methods*, Third Edition, 2002) and flexible probes. As we now note, this approach ensured consistency across the interviews, but also allowed the interviewer to follow up specific participant responses. In addition, in our description of the analysis we have alluded to (though not explored, because it is beyond the scope of the research questions addressed in this manuscript) unexpected findings in the data about the influence of broader socio-ecological factors on rural residents' adherence with diabetic eye screening, which we are analyzing in a separate study that we hope to publish.

12. Page 11 lines 15-45: We have revised our description of the analysis process to make it clear that the data analysis was iterative and developmental, requiring multiple cycles of coding and frequent discussions among the research team. We have also provided examples of the topics of these discussions, to give the flavor of the group analysis process.

13. Page 11 line 52: The quality improvement initiative began approximately 2 years after the establishment of the teleophthalmology program in which a group of clinic staff tested strategies for increasing utilization. This has been added to the Methods section.

14. Page 12 line 5: We fully agree and appreciate the Reviewer's comments about the importance of member-checking in light of our multiple points of engagement with participants in this study.

15. Page 12 line 38: All research activities in this study were reviewed by the University of Wisconsin-Madison Human Subjects Institutional Review Board (IRB) staff, which concluded that this study was exempt from full IRB review because it falls under U.S. federal Common Rule (45 CFR 46.101(b)), which considers interview research protocols with minimal risk to participants to be exempt. This has been clarified in the Methods.

16. Page 13 line 42-47: Primary care provider roles are now listed in full instead of appearing as acronyms in Table 1.

17. Page 14 Table 2: We agree that a fourth column with quotes would help enrich Table 2. However, we are limited by space and we found it impossible to incorporate a fourth column without making Table 2 very difficult to read. Therefore, we have elected to include the quotes in the text.

18. Page 14 line 47: We have clarified in our Results section that primary care providers were

motivated to use teleophthalmology because they believed that this technology made it easier for patients to obtain diabetic eye screening. In this community, patients must otherwise make their own appointments with an eye care provider for a dilated eye exam to obtain diabetic eye screening. This has been clarified in the discussion of primary care provider barriers and facilitators in the Results.

19. Pages 17-19: We appreciate Reviewer #1's comments that we have used a good selection of quotes to illustrate our points.

20. Page 19 lines 19-43: Figure 1 illustrates the temporal relationship between barriers in the teleophthalmology referral process, which is important for understanding the relative impact and mapping of possible strategies to overcome barriers in this process. This figure helps to demonstrate why strategies aimed at the provider and health system may be more effective in increasing teleophthalmology use because of the earlier role of the primary care provider and the later role of the patient in completing the referral process. This has been clarified in the text.

21. Page 21 line 21: This section has been renamed Discussion. Our data agree with prior studies in that a recommendation from their primary care provider is one of the strongest patient motivators to obtain preventative screening. We agree that patient-centered care and shared decision-making are important. However, it is clear that many patients choose to rely on their primary care provider to make their health care decisions because of the trust they place in their provider.

22. Page 22 line 13: It is now clarified in the Discussion that the organizational process of the clinic currently requires that the primary care provider provide a referral for a patient to obtain teleophthalmology. Some strategies identified to address this barrier include the delegation of the teleophthalmology referrals to clinic staff such as medical assistants or allowing patient self-referral. Diabetic eye screening is promoted during Diabetes Awareness Month (November) wherein patients are encouraged to obtain screening. However, this was not felt to be a significant facilitator by our participants.

23. Page 23 line 6: We have noted as a limitation of our study that we did not specifically assess participants' knowledge of diabetic eye screening in our study.

24. Page 24 line 8: Future research directions have been added to the Discussion.

25. Page 24 line 15: The Conclusion subheading has been changed to this paragraph.

Response to General Comment #2: Implications for other clinics include the importance of strategies targeting health system workflow processes, in addition to educating patients, providers, and staff. Clinics may benefit from assessing their own unique barriers and facilitators in order to tailor the selection of strategies to increase teleophthalmology utilization. These items have been added to the Discussion.

Reviewer #2

Response to General Comment: We appreciate Reviewer #2's extensive knowledge and experience with teleophthalmology programs. In our program, teleretinal imaging reports were returned within 7 days to patients and primary care providers (some in less than 24 hours). This was consistent with the usual timeframe for receiving results of other clinical studies (e.g. lab tests and x-rays) provided by this rural health system and was considered acceptable to all patients and primary care providers in our study. We have added this to our Methods. We agree that the optimal turnaround time is at the point-of-care with trained imagers performing the grading and providing results to patients at the time of imaging, particularly in low- and intermediate-income countries, facilitates scheduling referrals to

eye specialists.

1. More detail on the clinical and operational components of the teleophthalmology program have been added in the Methods section. The Topcon NW400 camera (Topcon Medical Systems, Inc., Oakland, NJ USA) was used to obtain a single 45-degree image of the disc and macula in each eye, along with an anterior segment photograph. If a fundus image was considered to be poor quality by the imager, then the camera's "small pupil" mode was used to capture additional images. If images remained poor, then pharmacologic dilation using 0.5% tropicamide was performed with the patient's consent. The percentage of patients undergoing pharmacologic pupil dilation was 2.2% and the frequency of ungradable cases among all patients was 2.6%.

2. Only one patient among the 20 patient participants underwent pupil dilation. There were no significant differences in the responses of this patient compared to those of the other 19 patients who did not undergo pupil dilation. This has been added to the Results section.

3. We appreciate Reviewer #2's suggestion of a mobile camera approach. A mobile camera approach was suggested by participants as one of several possible strategies to enhance patient convenience and has been added to the Results section.

4. We agree that the cost and availability of certain Electronic Health Record functionalities are a potential limitation for some health systems and that lower-cost alternatives or other implementation strategies may be more important in such settings. For example, one of the medical assistants devised a low-cost system of flagging paper charts with a color-coded diabetes checklist to make it easier for the PCP identify patients due for diabetic eye screening. This comment has been added in the Discussion section.

5. The rural population in our study faces limited access to care, which is shared by many non-Caucasian populations, including those in underserved urban areas and low- to medium-income countries. While we expect some of the implementation strategies we identified to translate to these populations, tailoring of strategies to the local community is important to account for differences in cultural backgrounds and available healthcare resources. This comment has been added in the Discussion section.

Again, we appreciate the Editor and Reviewers' insightful comments and suggestions. We believe that this manuscript is appropriate for publication by BMJ Open because it directly addresses research questions in clinical medicine relevant to public health, patient outcomes as well as the delivery of healthcare by improving our understanding of implementation strategies to address clinical barriers and facilitators to teleophthalmology use in rural U.S. multi-payer primary care settings with a goal of increasing access to diabetic eye screening and ultimately, reducing preventable blindness among patients with diabetes.

VERSION 2 – REVIEW

REVIEWER	Lil Deverell Swinburne University of Technology, Australia
REVIEW RETURNED	20-Oct-2018

GENERAL COMMENTS	Congratulations to the authors on your revision of this manuscript. The addition of definitions, explanations and local contextual details throughout means that your research story flows better, the connection between your actions and recommendations is more apparent and the report makes more sense to an outsider. I picked up a few copy-editing glitches, and was curious about one sentence,
--

	wanting more, but otherwise the article is very clear and well worth reading. Details as follows: Page 7 line 52 - missing a word "of": "...services through a variety of public or private..." Page 12 line 47 - The new statement "...findings included the influence of broader socio-ecological factors on rural residents' adherence..." made me very curious and while I understand you don't want to go into details in this article, it would be helpful to name a few examples in brackets of what you mean by socio-ecological factors (e.g., domestic violence, medical center was proximal to the betting shop, so health funds were spent on gambling not teleophthalmology). A few specific examples would provide a flag for subsequent researchers and health practitioners to consider how these factors might impact their own studies or review of medical service models. Page 24 line 5 - This sentence doesn't need the word "yet" in addition to "while" (line 3). Page 25 line 33 - missing word "to": "...make it easier for the PCP to identify patients..."
--	--

REVIEWER	Sven-Erik Bursell University of Hawaii, USA
REVIEW RETURNED	25-Oct-2018

GENERAL COMMENTS	The authors have provided appropriate responses to concerns and comments raised by this reviewer.
---